# Abundance of Poleroviruses within Tasmanian Pea Crops and Surrounding Weeds, and the Genetic Diversity of TuYV Isolates Found

**DOI:** 10.3390/v14081690

**Published:** 2022-07-30

**Authors:** Muhammad Umar, Robert S. Tegg, Tahir Farooq, Tamilarasan Thangavel, Calum R. Wilson

**Affiliations:** 1New Town Research Laboratories, Tasmanian Institute of Agriculture, University of Tasmania, 13 St. Johns Avenue, New Town, Hobart, TAS 7008, Australia; m.umar@utas.edu.au (M.U.); robert.tegg@utas.edu.au (R.S.T.); tamil.thangavel@daf.qld.gov.au (T.T.); 2Guangdong Provincial Key Laboratory of High Technology for Plant Protection, Plant Protection Research Institute, Guangdong Academy of Agricultural Sciences, Guangzhou 510640, China; tfarooq@gdppri.com; 3Department of Agriculture and Fisheries (Queensland), Bundaberg Research Facility, 49 Ashfield Road, Bundaberg, QLD 4670, Australia

**Keywords:** vegetables, legumes, polerovirus, *Turnip yellows virus*, NGS, phylogenetic analysis, genome diversity, recombination

## Abstract

The genus Polerovirus contains positive-sense, single-stranded RNA plant viruses that cause significant disease in many agricultural crops, including vegetable legumes. This study aimed to identify and determine the abundance of Polerovirus species present within Tasmanian pea crops and surrounding weeds that may act as virus reservoirs. We further sought to examine the genetic diversity of TuYV, the most commonly occurring polerovirus identified. Pea and weed samples were collected during 2019–2020 between October and January from thirty-four sites across three different regions (far northwest, north, and midlands) of Tasmania and tested by RT-PCR assay, with selected samples subject to next-generation sequencing. Results revealed that the presence of polerovirus infection and the prevalence of TuYV in both weeds and pea crops varied across the three Tasmanian cropping regions, with TuYV infection levels in pea crops ranging between 0 and 27.5% of tested plants. Overall, two species members from each genus, *Polerovirus* and *Potyvirus*, one member from each of *Luteovirus*, *Potexvirus*, and *Carlavirus*, and an unclassified virus from the family *Partitiviridae* were also found as a result of NGS data analysis. Analysis of gene sequences of the *P0* and *P3* genes of Tasmanian TuYV isolates revealed substantial genetic diversity within the collection, with a few isolates appearing more closely aligned with BrYV isolates. Questions remain around the differentiation of TuYV and BrYV species. Phylogenetic inconsistency in the P0 and P3 ORFs supports the concept that recombination may have played a role in TuYV evolution in Tasmania. Results of the evolutionary analysis showed that the selection pressure was higher in the *P0* gene than in the *P3* gene, and the majority of the codons for each gene are evolving under purifying selection. Future full genome-based analyses of the genetic variations will expand our understanding of the evolutionary patterns existing among TuYV populations in Tasmania.

## 1. Introduction

Vegetables are a major dietary source of many essential nutrients and fibre in a balanced human diet. The Australian vegetable industry is a significant part of the national economy, providing both fresh and processed vegetables and vegetable products for domestic consumption and for export. With a gross value of $4.1 billion, vegetable production was Australia’s sixth highest-value agricultural industry, with exports accounting for $385 million of the country’s agricultural export revenue in 2017–18 [1]. In Tasmania, the vegetable sector contributes substantially to the state’s economy, with a gross value of $299 million in 2019–20 and with up to $21 million of export revenue [2]. However, numerous plant pathogens, including viruses, can infect vegetables, posing a significant threat to crop growth, yield, quality, and market value.

Green peas are grown on significant areas of Tasmanian land and are known to host a variety of viruses from various families, including *Luteoviridae*, *Solemoviridae*, *Potyviridae* and others (Table 1) [3,4]. Chlorosis, vein clearing, mottling, dwarfing, enations, and necrosis are some of the symptoms caused by these viruses in single or mixed infections [5,6,7,8,9]. Among the viruses most commonly found infecting green pea in Tasmania are members of the *Polerovirus* genus, in particular *Turnip yellows virus* (TuYV) [3]. The *Polerovirus* genus (family *Solemoviridae*) is genetically diverse, with 32 approved or tentative species [10] that share biological characteristics such as persistent transmission by insect vectors (primarily aphids) and phloem-limitation in the plant hosts [11,12,13]. Examination of virus abundance, distribution, diversity, and alternative hosts will be critical for developing appropriate management strategies [3,14].

TuYV was first reported in the United Kingdom as a European strain of *Beet western yellows virus* (BWYV) based on biology and serology [15]. TuYV and BWYV were later classified as separate species by the International Committee on Taxonomy of Viruses (ICTV) based on different host ranges [16]. TuYV has a monopartite linear single-stranded RNA (ssRNA) genome that contains seven overlapping open reading frames (ORF0 to ORF5 and ORF3a). These ORFs encode proteins associated with different functions, including suppression of the host silencing response and virus accumulation (P0), virus replication (P1–P2), systemic infection and phloem-limitation (P3a and P4) and capsid proteins and vector recognition (P3 and P3–P5) [17,18,19,20,21,22,23]. P3 is the major coat protein (CP) essential for RNA stability and virion assembly. The P3–P5 fusion of CP with the P5 readthrough domain (RTD) is a minor component of the capsid. The RTD, exposed on the virion surface, is not required for infection initiation or virion assembly but is necessary for vector transmission and virus circulation in plants [23,24].

In general, the *Polerovirus* genus is poorly understood in terms of variation and conservation. A number of proteins within the genus have been identified as multifunctional proteins that are typically highly disordered and hypervariable [25]. It has been demonstrated that P0 and the RTD of P3–P5 appear to be hypervariable regions and are likely viral species determinants, whereas P2 and P3 are the most genetically stable cistrons in the *Polerovirus* genome.

The error-prone nature of RNA-dependent RNA polymerase (RdRp) creates a significant potential for genetic variation in RNA plant viruses, including TuYV [11,26]. Mutation and recombination are the two most common types of errors that are thought to cause genetic variation. Evolutionary factors such as genetic drift and selection, including selection pressures associated with virus-vector selection, host plant selection, and maintaining functional structures, could all influence genetic variation [27]. Several molecular studies have demonstrated significant levels of genetic diversity within TuYV isolates infecting different crops, including oilseed rape, pulses, beetroot etc. [3,14,28,29,30]. Furthermore, P0 and P3 play a key role in TuYV virulence [17,31]. Therefore, the mode of evolution of these cistrons can reveal information about the epidemiological dynamics of TuYV.

TuYV has a wide host range that includes numerous crops and weed species from different plant families like *Brassicaceae*, *Fabaceae*, *Chenopodiaceae*, and *Compositae* [15,32,33,34,35,36]. The wide host range increases the potential reservoirs of TuYV inoculum and acts as a “green bridge” for both virus and its aphid vectors [37]. The diverse range of hosts and vectors are the main factors aiding the worldwide distribution of TuYV. Transmission of TuYV occurs by aphids in a persistent, circulative manner, with varying degrees of efficiency [38,39]. *Myzus persicae* (green peach aphid) is often considered the most important vector due to its abundance, wide host range and transmission efficiency [39].

The objective of this study was to determine the prevalence, incidence, and genetic diversity of polerovirus species associated with pea crops and surrounding putative weed or pasture virus reservoir hosts in Tasmania. The outcome of this study will provide Tasmanian pea growers with a clearer understanding of what viruses may pose a threat to their crops, their putative inoculum sources and whether those potential pathogens may require management.

## 2. Materials and Methods

### 2.1. Field Survey and Sampling

Leaf samples from green pea plants and weed species found in the proximity of these pea fields were collected during the cropping period from October 2019 to January 2020 from commercial pea crops in three different regions (far northwest, northwest, and central north) of Tasmania (Figure 1A). For weed sampling, 36 sites were visited at or just prior to planting of the pea crop, and at least ten samples were collected for each species. Overall, 820 weed samples were collected with 3–5 different weed species per site. Pea leaf samples were collected at crop maturity (between the 10th and 12th week from planting) from 22 of the 36 sites, with 100 leaf samples taken at random per site. All leaf samples (weeds and pea) were grouped into lots of five prior to virus testing. Details of the types and numbers of collected weed samples are summarized (Figure 1B). The metadata were recorded, including coordinates of sites, sample, season, plant host, symptoms, and the average seasonal temperature for that region (Appendix A). Samples were stored frozen at −80 °C until subsequent RT-PCR testing.

### 2.2. RNA Extraction and Detection of Poleroviruses through RT-PCR

Total RNA was extracted from 100 mg of each 5-leaf lot (to ensure sufficient total RNA yield for further processing, thus allowing detection of low-titre viruses) using a PowerLyzer 24 homogeniser (QIAGEN, Chadstone, VIC, Australia) and RNeasy plant mini kit (Qiagen, Hilden, Germany) following the manufacturer’s protocol. cDNA was synthesised using the iScript Reverse Transcription Supermix Kit (BioRad Bio-Rad, Hercules, CA, USA) following the manufacturer’s protocol in a 20 μL reaction volume containing 2.5 μL total RNA, 4 μL of 5× RT Supermix made up to volume with sterile distilled, and DEPC (diethyl pyrocarbonate)-treated water. The reaction was incubated at 22 °C for 5 min (priming) and 46 °C for 20 min (reverse transcription), followed by enzyme inactivation at 95 °C for 1 min.

Initially, the samples were screened by PCR using universal *Luteoviridae* primers (C2F1 + C2F2 and C2R1 + 2) [40], and any positive samples were selected and screened further using species-specific primer sets for TuYV [41], BWYV [3] and *Phasey bean mild yellows virus* (PBMYV) [42,43] (Table 2). Details, including primer sequences, amplicon size, target, and thermocycling conditions, are given in Table 2. All assays were performed in the following reaction mix: 1× HotStarTaq Master Mix (Qiagen), 10 μM of each primer pair, 1.75 μL cDNA template and made up to a total volume of 20 μL with sterile distilled water. PCR products were visualised by electrophoresis in a 1% agarose gel stained with SYBR Safe DNA Gel Stain (Invitrogen) in 1× lithium borate buffer, using a 1-kb and 100-bp molecular weight marker (Genedirex).

Amplicons of the appropriate sizes were then excised and purified using a QIAquick Gel Extraction Kit Protocol (Qiagen). Elution was done using 30 μL elution buffer (10 mM Tris HCl, pH 8.5) and sent for bidirectional sequencing by the Australian Genome Research Facility (AGRF) and Central Science Laboratories (CSL), University of Tasmania, Australia. A further 51 luteovirid-positive PCR products of unknown isolates (which showed negative results with species-specific primers) were also purified and sequenced.

### 2.3. Next-Generation Sequencing (NGS)

Leaf samples of different plants of the same species, representing all three Tasmanian regions, were pooled into six samples (Appendix A). The total RNA of these six samples was extracted using an RNeasy plant mini kit and sent to the Department of Jobs, Precincts and Regions, AgriBio, Bundoora, Victoria for NGS. RNA-Seq libraries were prepared using a TruSeq Stranded Total RNA Sample Preparation kit with Ribozero, quantified using qubit and 2200 TapeStation system (Agilent Technologies, Santa Clara, CA, USA) and run-on Illumina NovaSeq with a paired read length of 2 × 151 bp. Trimming of adaptor and primer sequences from the reads was carried out using the BBDuK plugin package [44] in Geneious Prime 2021.2 (Biomatters Ltd., Auckland, New Zealand). Following that, de novo assembly was done using the Velvet plugin implemented in Geneious Prime with default settings. The resulting contigs were then sorted by length and blasted (BLASTn and BLASTx) against GenBank database. Simultaneously, the mapping of contigs and reads to a collection of polerovirus reference genomes (available in GenBank) was carried out, and results were also analysed in Geneious Prime. For validation, the NGS sequences were also analysed using Virfind, a virus detection pipeline [45], to screen for all possible viruses in the databases.

### 2.4. Data Analysis and Phylogenetic Tree Analysis

The number of single and multiple virus infections was determined and tabulated for tested samples from all locations. The virus incidence within each crop was estimated from grouped samples [46].

All the nucleotide sequences were edited and analysed with Geneious Prime and compared to the sequences already published in the NCBI GenBank database using BLASTn and BLASTx. To prepare multiple sequence alignments, complete nucleotide sequences of both P3 and P0 of all known poleroviruses isolates as well as of all available TuYV and BrYV isolates were retrieved from GenBank and aligned with the sequences obtained in this study (Appendix A) using the MUSCLE option in Geneious Prime. All alignments were manually reviewed and adjusted as needed before being subjected to further analyses.

The phylogenetic trees were constructed in MEGAX using the maximum likelihood method with 1000 bootstrap values [47]. Visualisation and editing of phylogenetic trees were done using iToL [48]. Finally, sequence identity among TuYV isolates was determined using Geneious Prime.

### 2.5. Recombination Analysis

The sequence datasets of P0 and P3 were individually analysed to determine evidence for any recombination events. The recombination analysis was carried out using RDP [49], GENECONV [50], BOOTSCAN [51], MAXCHI [52], CHIMAERA [53], SISCAN [54] and 3SEQ [55] methods implemented in the recombination detection program v.4 [56]. Aligned sequences were subjected to the above-mentioned methods using default parameters. *p*-values below a Bonferroni-corrected cutoff of 0.05 were considered statistically significant. The recombination signals detected by at least three different methods were considered reliable.

### 2.6. Nucleotide Diversity and Haplotype Variability Indices

The average pairwise number of nucleotide differences per site (nucleotide diversity, π) was estimated for all samples of each gene (*P0* and *P3*) using DnaSP v.5 [57]. The statistically significant differences in the mean nucleotide diversity from both datasets were determined by calculating their 95% bootstrap confidence intervals. In addition, the nucleotide diversity was determined using a 100-nucleotide sliding window with a 10 nucleotide step size. The number of segregating sites (S), the number of haplotypes (H), and haplotype diversity (Hd) were also calculated for all datasets using DnaSP v.5 [58].

### 2.7. Detection of Sites under Positive and Negative Selection Pressure

Using four different approaches, potential negatively and positively selected sites in the P0 and P3 coding regions were identified. These were single-likelihood ancestor counting (SLAC), fixed-effects likelihood, random-effects likelihood, and partitioning for robust inference of selection [59]. All four analyses were conducted on the Datamonkey web server (www.datamonkey.org, accessed on 10 October 2021) [60]. Genetic algorithm recombination detection (GARD) [61] was implemented to look for recombination breakpoints in datasets in order to avoid ambiguous results. The dN/dS ratios were estimated using the SLAC method based on inferred GARD-corrected phylogenetic trees to compare the selection pressures acting on the *P0* and *P3* genes of TuYV.

## 3. Results

### 3.1. Virus Incidence and Prevalence

Although no obvious visual symptoms associated with virus infection were noted during sample collection, PCR results showed that luteovirid infections were prevalent both in weeds and pea crops across the different locations surveyed (Table 3 and Table 4). Notably, TuYV was an abundantly detected virus amongst the tested plants. RT-PCR results indicated that out of 164 weed samples, 66 were found to be positive when tested with Luteoviridae universal primers. Out of 66 *Luteoviridae* positive weed samples, 29 tested positive with the TuYV species-specific primers.

Likewise, luteovirid infections were also recorded in 151 of 440 pea samples tested by PCR. Of those, 137 pea samples tested positive with the TuYV species-specific primers. Notably, none of the weed and pea samples were detected by BWYV and PBMYV-specific primers (Table 3). Additionally, 37 weeds and 14 peas luteovirid positive samples were not amplified by the species-specific PCRs used. Subsequent sequence analysis (of these luteovirid PCR products) using Blastn showed that 24 (weeds) and 5 (pea) were infected by the *Soybean dwarf virus* (SbDV). However, the nucleotide sequences of 13 (weeds) and 9 (pea) remain undetermined. Overall, TuYV infection was found in 18 of 22 pea crops, with incidences ranging from 2.1% (in the Fairfield district) to 27.5% (in the Palmerston region) (Table 4).

### 3.2. NGS of Selected Samples

The total numbers of raw reads obtained with Illumina sequencing for the six samples, U1, U2, U3, U4, U5, and U6 were 22,351,000, 16,807,356, 18,774,804, 17,655,072, 17,311,114, and 19,718,506, respectively. After quality trimming using BBDuK, these numbers were reduced to 22,228,228, 16,721,940, 18,745,222, 17,604,404, 17,257,356, and 19,933,492, respectively. These remaining reads were used to generate Polerovirus contig sequences by de novo assembly and mapping against 131 reference sequences of Polerovirus genomes. For sample U1-U6, 670, 436, 716, 796, 262 and 9349 reads produced 16, 12, 10, 16, 16, and 22 Polerovirus contigs, respectively. A BLASTn/BLASTx search of GenBank databases revealed the existence of contigs corresponding to TuYV, *Brassica yellows virus* (BrYV), *Turnip mosaic virus* (TuMV), *Raphanus sativus cryptic virus* (RsCV), SbDV, *White clover mosaic virus* (WClMV), *Red clover vein mosaic virus* (RCVMV), and *Clover yellow vein virus* (ClYVV) (Table 5). For TuYV, TuMV, RsCV, SbDV, RCVMV, and ClYVV, the obtained contigs covered the complete/near-complete genome of each virus, while for BrYV and WClMV, contigs covered 70.4 and 28.9% of their total genomes, respectively (Table 5). The sequence of the BrYV isolates (accession no: OM469309) was validated by RT-PCR, followed by Sanger sequencing of the amplicons obtained [62]. Other sequences remain tentative until further validation.

### 3.3. Nucleotide Sequence and Phylogenetic Analysis of Isolates Testing Positive by TuYV Species-Specieis PCR

#### 3.3.1. P0 of Tasmanian Isolates with P0 of All Reported Poleroviruses

The P0-based phylogenetic analysis of Tasmanian isolates with globally reported *Polerovirus* species revealed that the overall nucleotide sequence identity ranged between 21.4 and 100%. Tasmanian TuYV isolates shared close sequence similarity with a published strain of TuYV (NC_003743; 84.3–94.0%) and BrYV (NC_016038; 82.5–96.1%) but not with any other polerovirus species tested (Appendix A).

#### 3.3.2. P0 of Tasmanian Isolates with P0 of Worldwide Reported TuYV and BrYV Isolates

Given the possible association of some isolates with the BrYV P0 sequence, we realigned the 75 P0 sequences of Tasmanian isolates from this study with 66 TuYV and 19 BrYV isolates retrieved from GenBank. The overall percentage of sequence identities ranged between 83.3 and 100% (Appendix A).

The phylogenetic analysis based on the *P0* gene of Tasmanian and worldwide reported isolates revealed the formation of 11 distinct groups based on the country of their origin (Figure 2). These groups further subdivided the TuYV and BrYV P0 populations based on the host plant they infect. For instance, a majority (74 isolates) of the Australian TuYV population (including the isolates from this study) were found to be associated with pea, followed by *B. napus* (16 isolates), *R. raphanistrum* (12 isolates), *T. repens* (6 isolates), *Arctotheca calendula* (4 isolates), *Cicer arietinum* (4 isolates) and *Lens culinaris* (2 isolates). As for other host plants such as *Beta vulgaris*, *Diuris* sp., *Sinapis arvensis*, *Trifolium fragiferum*, *Trigonella foenum* and *Vicia sativa*, each host was associated with one isolate (Figure 2). The Serbian population (six isolates) was subdivided into four groups and were reported from *B. napus* (two isolates), *Brassica oleracea* (two isolates), *B. nigra* (one isolate), and *Sinapis alba* (one isolate). Four TuYV isolates from Greece came from *B. napus*, while three isolates from China originated from *B. pekinensis*, *Raphanus sativus*, and *N. tabacum*. Two isolates from Iran were from *B. napus* and *Sinapis arvensis*, while the only isolates from Germany and Poland were from *Physalis pubescens* and *B. napus*, respectively. The only isolate from France was from *Lactuca sativa*. The origin of two isolates remained unknown due to a lack of host plant information. Similarly, the BrYV population from China (ten isolates) was found to be associated with *B. napus* (four isolates), followed by *Brassica campestris* (two isolates), *Nicotiana tabacum* (two isolates), *R. raphanistrum* (one isolate), and *Brassica rapa* (one isolate). Eight BrYV isolates from Japan were subdivided into five groups and were reported from *B. napus* (three isolates), *B. rapa* (two isolates), *Sinapis alba* (one isolate), *B. oleracea* (two isolates) and *B. napus* (two isolates). The only South Korean isolate was from *B. rapa.*

The phylogenetic analysis tree separated all the isolates into three major clades, A, B, and C (excluding the outgroup). Clade A included six already-reported TuYV isolates from different countries (four from Serbia and one each from Greece and Poland). Clade B includes all 19 BrYV (10 from China, 8 from Japan, and 1 from Korea) isolates along with 5 Tasmanian isolates from this study and 19 previously reported isolates (12 from Australia, 3 from China, and 2 from Serbia) (Figure 2). A total of 7 isolates out of these 12 Australian isolates were reported from Tasmania previously [3]. Clade B could be further subdivided into two clades (B1 and B2), with 16 of 19 BrYV worldwide isolates residing in B1 with three previously reported TuYV isolates. Five Tasmanian isolates were present in clade B2 with the three previously reported BrYV and 16 TuYV isolates. Clade C included 111 TuYV isolates in total, which could be further subdivided into seven sub-clades C1, C2, C3, C4, C5, C6 and C7. Clade C1 included five Tasmanian isolates from this study and a previously reported Australian isolate. Another five Tasmanian isolates (this study) clustered with three previously reported isolates from Australia in clade C2. Clade C3 includes 9 Tasmanian isolates, along with 12 Australian, 2 Iranian and 2 Greek isolate. A further ten Tasmanian isolates from this study clustered with another one previously reported isolate from Australia in clade C4. Clade C5 included 14 Tasmanian (this study) and 7 previously reported Australian isolates. Another nine Tasmanian isolates were residing in sub-clade C6. Finally, all remaining 17 Tasmanian isolates were grouped into sub-clade C7 along with 11 Australian and one each from Greek and French isolate (Figure 2).

#### 3.3.3. P3 of Tasmanian Isolates with P3 of All Reported Poleroviruses

The sequence analysis of the *P3* gene of Tasmanian isolates with the *P3* gene of all reported *Polerovirus* species revealed that diversity within Tasmanian isolates ranged from 91.2 to 100%. Comparisons to the other polerovirus species showed Tasmanian TuYV isolates were closely related to TuYV (NC_003743), BrYV (NC_016038), BWYV (NC_004756), *Faba bean polerovirus 1* (FBPV-1) (NC_055495), (*Beet mild yellowing virus*) BMYV (NC_003491), and *Beet chlorosis virus* (BChV) (NC_002766), sharing a percentage of sequence similarities ranging from 90.7 to 98.2% (Appendix A). The phylogenetic analysis separated the Tasmanian and worldwide isolates into three major clades, A, B, and C, with all the Tasmanian isolates residing with the six closely related viruses (mentioned above) in clade C (Appendix A).

#### 3.3.4. P3 of Tasmanian Isolates with P3 of Worldwide Reported TuYV and BrYV Isolates

The P0 analyses suggested Tasmanian isolates are unrelated to BWYV, FBPV-1, BMYV and BChV but may share some similarities to BrYV, and so P3 data were realigned and analyzed with *P3* genes of 86 TuYV and 19 BrYV isolates retrieved from GenBank, revealing significant genomic diversity with an overall sequence identity ranging from 88.1–100% (Appendix A).

The phylogenetic analysis based on the *P3* gene of Tasmanian and worldwide reported isolates divided a total of 130 Australian isolates (including from this study) into 14 unique groups. Of these, 67 TuYV isolates originated from pea, followed by *B. napus* (16), *B. oleracea* (14), *R. raphanistrum* (12), *T. repens* (6), *A. calendula* (4), *Cicer arietinum* (4), *Lens culinaris* (2), *Beta vulgaris* (1), *Diuris* (1), *Sinapis arvensis* (1), *Trifolium fragiferum* (1), and *Vicia sativa* (1). Of the 15 German isolates, 13 were associated with pea. *Physalis pubescens* and *Raphanus raphanistrum*, were associated with one isolate each. Three TuYV isolates from Iran were from *Medicago sativa*, while two isolates from Colombia originated from *Arachis pitoi*. Two isolates from Poland were from *B. napus*, while two isolates from Pakistan originated from *B. oleracea* and *Spinacia oleracea*. The only isolates from Egypt and Morocco were from *Vicia faba*, and the only isolates from China and the United Kingdom were from *Nicotiana tabacum* and *B*. *napus*, respectively (Figure 3). The origin of the two isolates remained unknown due to the lack of information on the host plant.

Further analysis showed that the tree separated all the isolates into eight major clades (A to H) (excluding the outgroup). Clade A included only one Tasmanian isolate (92P), whereas seven TuYV (two each from Australia, Pakistan, and France and one from China) isolates clustered together with six BrYV (four from China and two from Japan) in clade B. Clade C contained 19 Tasmanian (this study), 8 Australian, 6 German, and 1 isolate from Poland, and 1 South Korean BrYV isolate. A total of 4 Tasmanian isolates shared clade E with 16 Australian isolates. Clade F contained 10 Tasmanian isolates, sharing this clade with 11 previously reported isolates from Australia, followed by nine from Germany, three from Iran and each isolate from Egypt, Morocco, Poland, and the UK. However, two isolates from this clade have missing information about their origin and host. Thirty Tasmanian isolates clustered with three already reported Australian TuYV isolates in clade G. Clade H included 14 Tasmanian isolates. Finally, all remaining 10 Tasmanian isolates clustered with 11 Australian and 2 Colombian isolates (Figure 3). As has been previously shown, P3 sequence analyses did not discriminate between TuYV and BrYV isolates.

### 3.4. Recombination Analysis

The phylogenetic analysis showed that the clustering of Tasmanian TuYV isolates varied between P0 and P3 and that both trees were not congruent. These findings point to possible recombination events in the P0 and P3 of several Tasmanian TuYV isolates.

The recombination analysis results suggested that the 36W isolate was a recombinant of haplotypes represented by the isolates 225P and 345P (*p* ≤ 7.386 × 10^−3^), detected within the *P0* gene. One event of recombination within *P3* genes was detected, indicating 88P as a (putative or moderate) recombinant of haplotype represented by the isolates 92P and 162W (*p* ≤ 3.451 × 10^−7^) (Table 6).

The analyses also revealed the recombination breakpoints i.e., putative recombination hot spots within the *P0* gene (in five sequences), at positions 32–390 (without gaps) in the alignment (corresponding to nucleotides 34–409 of the *P0* gene) (Figure 4A). The recombination hotspots within the *P3* gene were detected (in 34 sequences) at positions 23–310 (without gaps) in the alignment (corresponding to nucleotide 24–312 within the gene) (Figure 4B).

### 3.5. Nucleotide Diversity and Haplotype Variability Indices

The analysis of genetic diversity within the *P0* and the *P3* genes confirmed that both genes were variable with a high number of mutations, a high number of polymorphic sites and very high haplotype diversity but low nucleotide diversity. However, the *P0* gene had greater genetic diversity as compared to the *P3* gene (Figure 5 and Table 7).

### 3.6. Detection of Sites under Positive and Negative Selection Pressure

In order to gain a better understanding of the potential role of selection pressure on the genomic variation observed between analysed P0 and P3 datasets, the non-synonymous to synonymous substitutions (dN/dS) for each gene were compared. The average dN/dS ratio for both genes (*P0* and *P3*) remained low, i.e., dN/dS < 1, implying that the observed genomic variation on both P0 and P3 is being driven by negative selection (Table 8). The average dN/dS ratio was 0.6198 and 0.4245 for the P0 and P3, respectively, which showed the selection pressure was higher in the *P0* gene than in the *P3* gene (Figure 5A; Table 8). A total of 201/222 sites in the *P0* gene sequence were found to be under negative selection pressure, with the values ranging between 0.039 and 0.982. Likewise, 145/158 negatively selected sites for the *P3* gene were found, with values ranging between 0.056 and 0993. Interestingly, a few sites under positive selection pressure were also detected, though the proportion remained lower (20/222 for *P0* and 12/158 for the *P3* gene) than that of sites under negative selection pressure (Figure 5B; Table 8).

## 4. Discussion

In this study, we undertook a comprehensive survey of Tasmanian pea crops and neighbouring weed populations that may be potential virus reservoirs using RT-PCR and NGS approaches. Our findings will facilitate a broader understanding of the distribution, diversity and genomic variability of viral populations associated with pea cropping in Tasmania.

We showed that TuYV is the most prevalent and widely distributed Polerovirus in all surveyed regions, with virus infection levels in crops ranging between 0 and 27.5% of tested plants. The virus was found in 18 of the 22 pea fields located in all three cropping regions. This finding corroborates an earlier report of TuYV incidence in Tasmanian pea crops [3], where polerovirus (predominantly TuYV) infections were detected in pea crops across three seasons. However, in contrast to the data presented in this manuscript, Polerovirus prevalence did not exceed 60% of surveyed crops, and incidence did not exceed 6.7% in any field in the prior survey. Within the weed hosts neighbouring pea crops, we found that the TuYV infections were greater in *R. raphanistrum* and *A. calendula*, followed by *T. repens*, *Vicia sativa*, *Trifolium fragiferum*, and *Sonchus* sp. TuYV is known to have a wide host range that includes weeds, legume pasture species and other plants from the families *Brassicaceae*, *Fabaceae*, *Amaranthaceae* and *Asteraceae* [14,33]. It has also been reported across Australia to infect pulse crops (including peas), canola and various other weed species on a regular basis [14,36]. A further distinction from the previous Tasmanian survey was a failure to confirm the presence of the newly described PBMYV [63] in Tasmanian pea crops, which was previously found at low infection levels corresponding to 9 of 28 polerovirus infections detected [3].

The rate of TuYV infection varied according to the type of host plant and the location of the sampling sites. However, sampling bias must be considered when discussing the distribution of TuYV and the variation in its infection rate across plant species. We hypothesise that several other factors might also be involved in the variable incidence of the virus, such as stage of infection (early or late), types of aphid vectors and their population dynamics, crop management practices, climatic conditions, etc. However, additional studies are imperative to better understand the specific impact of these factors on TuYV incidence.

In addition, our NGS data confirmed several other viral reads, including an almost full genome of viruses representing five different genera and one unassigned virus. The genus Polerovirus and Potyvirus were each represented by two species members, followed by one member from each of *Luteovirus*, *Potexvirus*, and *Carlavirus*. The unclassified virus was from the family *Partitiviridae* (Table 5). However, except for BrYV, RT-PCR was not performed to further confirm the presence of the viruses detected by NGS [62]. Notably, three of these viruses (BrYV, RCVMV and RsCV) represent previously undescribed viruses in Tasmania [4].

BrYV belongs to the genus *Polerovirus* and is closely related to but appears to be distinctive from TuYV in terms of the P0 and P5 cistrons sequences [64,65,66]. It has been reported in Asia [64,65,66], and a recent report from Australia described the concatenated ORF-based phylogenetic analysis of collected isolates and demonstrated the paraphyletic relation between TuYV and BrYV [14]. The host range of BrYV is similar to that of TuYV [67]. However, BrYV isolates have also been subdivided into three genotypes (BrYV-A, -B, and -C) based on sequence similarity and phylogeny, with greater divergence seen in P0, P1 and P2 than in P3, P4 and P5 [65,66].

RCVMV is aphid-transmissible in a non-persistent manner; it can also be transmitted mechanically and by seed [68,69,70,71,72]. It was first reported in the United States (USA) [70] and then more widely [73,74,75,76], most recently in New Zealand [77]. Depending on the host species and virus isolate, RCVMV can cause vein mosaic, vein chlorosis, and plant stunting, as well as latent infections [73,76,78,79]. Though RCVMV has been shown to cause leaf chlorosis and yield loss in susceptible varieties of *Trifolium pratense* [80], assessment of the potential impact of RCVMV on clover requires further studies. That is because other viruses (e.g., AMV, *Bean yellow mosaic virus* (BYMV), ClYVV, or WClMV, all present in Tasmania) also naturally infect clover and are often found in mixed infections [77,81]. Although the presence of RCVMV in Tasmania has been confirmed (unpublished data), future surveys of legume and pasture crops should be conducted to determine its prevalence.

RsCV is an unclassified member of the family *Partitiviridae*, and members of this family are known to infect their hosts asymptomatically, including fungi, plants, some protozoa, and possibly some higher animals [82,83,84]. Members have two essential, double-stranded RNA genome segments ranging from 1.4 to 3.0 kb with RNA1 encoding RdRp, while RNA2 encodes CP [82]. Among those, plant-infecting partitiviruses are commonly referred to as cryptoviruses. Their host range includes radish, alfalfa, beet, broad bean, carrot, *Brassica* spp., white clover, red clover, rose, carnation, hop trefoil, Italian ryegrass, meadow fescue, spinach pear, and pine [84]. They are seed-transmitted with high efficiency, but there is no systemic infection or cell-to-cell moment due to the lack of a movement protein, so they move vertically through their host’s cell division processes instead [85].

In order to determine the genetic variability of Tasmanian isolates identified by TuYV species-specific PCR, two distinct segments in the genome were sequenced: ORF0, which codes for a protein (P0) involved in post-transcriptional gene silencing, symptom expression, and host range specificity [86,87], and ORF3, which codes for the coat protein (P3) [88,89]. Seventy-five Tasmanian isolates shared nucleotide identities ranging from 83.2–100% for P0 and from 91.2–100% for P3. Furthermore, the P0 and P3 sequences of the Tasmanian isolates shared 84.6–100% and 89.3–100% nucleotide identities with already published TuYV isolates from GenBank. These results are consistent with previous studies of genetic variation in the *P0* and *P3* genes from the UK [26,87]. One of the findings showed that the isolates shared 91.7–100% nucleotide identity for P0 and 94–100% for the P3 gene. The range of nucleotide identities was 86.9–98.8% and 93.5–99.8% for the *P0* and *P3* genes, respectively, when compared with the published isolates from GenBank [26]. In another study, the European TuYV isolates shared 81.1–100% of the *P0* gene and 90.6–100% nucleotide identities [87]. The genetic variation of these ORFs indicated that the genomes of polerovirus within these regions are more diverse than previously assumed. When isolates identified as BrYV were included in the phylogenies, we found evidence for discrimination of TuYV and BrYV isolates within the *P0* gene sequences, but no such clear differentiation was found with P3 sequences. Of the new Tasmanian isolates tested, five appeared to cluster with BrYV rather than TuYV in the P0 analysis and may be considered variants of this virus. Notably, subsequent testing of these variant isolates by BrYV species-specific PCR [90] did not result in positive amplification. Whilst evidence for the separation of BrYV and TuYV as distinct species exists from this and prior studies [62,64,65,66], suggestions that these viruses may represent a single highly variable species have been made [14]. Further whole genome sequencing of diverse isolates would be valuable to test these two propositions.

Phylogenetic analysis also highlighted similarities in sequence identity between isolates obtained from weed plants in the vicinity of pea crops and those found within the crops themselves (Figure 2 and Figure 3), pointing to the likelihood of virus exchanges between these wild and agricultural plant populations. Additional experimental analyses, including vector and virus monitoring, are required to determine the transmission rate and pattern.

Recombination is more common among RNA viruses than DNA and is a major driving force that greatly contributes to the evolution of viral populations [91]. Recombination governs genomic diversity and facilitates viral adaptation to varying environments (new host and environmental adaptation), ultimately resulting in the emergence of new/resistant-breaking/virulent variants or strains [92,93]. As previously proposed, poleroviruses tend to show higher recombination rates that further contribute to the emergence of new species and their evolution [94]. The evolution of the genus *Polerovirus* is marked by both intraspecific, homologous and interspecific, non-homologous recombination [14,95], and generally, in *Luteoviridae*, the recombination breakpoints are often at the boundaries of the gene rather than within the gene [96]. This implies that recombination events in the genome do not occur randomly; rather, these are associated with specific hotspots in the viral genome. Our findings suggested that recombination could be important and may play an important role in the evolution of these viruses. Interestingly, one isolate identified as a BrYV variant based on P0 sequence analysis was subsequently identified as a putative recombinant isolate in recombination analysis. This may suggest genetic exchange between these viruses species may be occurring. We found that recombination breakpoints were detected among 5/75 and 34/75 sequences of *P0* and *P3* genes, respectively (Table 6), which strongly suggests that the *P0* and *P3* genes have distinct evolutionary histories. Additionally, the considerable phylogenetic incongruence in *P0* and *P3* found in our work supports the concept that recombination might have played a role in the evolution of TuYV. Notably, recombination in genes or mutation in these proteins can affect the biological functions of the viral proteins. For instance, previous studies suggest that host range of different TuYV isolates may be influenced by genetic variation within the *P0* gene [26,87]. On the other hand, P3 has a biologically active region that plays a vital role in CP subunit interactions, plant–virus interactions and aphid–virus recognition. Furthermore, the viral particle’s assembly is required for vector transmission and determines which insect vectors can transmit the virus [97,98,99]. Thus, recombination and mutation might affect critical biological functions governed by the *P3* gene. The *P5* readthrough component of the capsid protein has been shown to be highly variable within sequenced TuYV species and across polerovirus species [87,100]. The biological significance of variation in the *P5* gene remains unclear at this stage but could be associated with specificity to different vector species. We did not include the analyses of the *P5* gene in this study, which should be considered in future research to acquire a more in-depth understanding of the recombination-driven genome variation.

Our results showed that both *P0* and P3 are primarily evolving under purifying selection pressure. Our findings revealed that with an average dN/dS ratio of <1, the majority of the codons remained under negative selection (Figure 5 and Table 8), and the overall contribution of negatively selected sites remained >90% (91% for *P0* and 92.4% for *P3*). This is consistent with the previous research that concluded that both genes of TuYV (*P0* and *P3*) were evolving under strong negative selection pressure [26]. Negative selection in the TuYV genome, which is required to keep the encoded protein functional (as in P3 and P0 of TuYV), may have helped to eliminate deleterious variants. TuYV’s *P0* gene is involved in RNA-silencing suppression [17], and mutations in this gene are expected to pose a significant impact on virus fitness, limiting genetic diversity and thus influencing dN/dS ratio estimates [101,102].

Taken together, this study advances our current understanding of the genetically diversified and evolving populations of TuYV infecting Tasmanian pea crops. Our findings also provide a practical framework necessary for understanding the current diversity and distribution of poleroviruses in three selected regions of Tasmania. It also provides important information on epidemiological aspects and management of viral diseases of pea crops. Future studies based on the full genome-based analyses of the genetic variations will expand our understanding of the evolutionary patterns existing among TuYV populations in Tasmania.

## Figures and Tables

**Figure 1 viruses-14-01690-f001:**
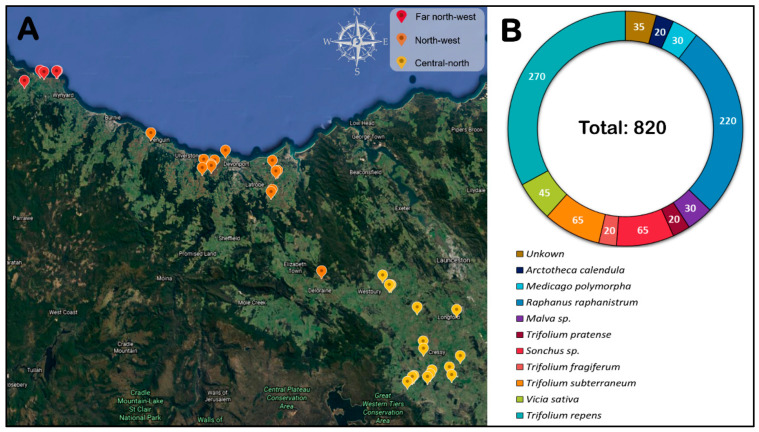
(**A**) Map of the three different pea crop sampling regions (far northwest, northwest, and central north) in Tasmania. (**B**) Details of weeds and pasture species samples collected from all sites prior to pea crop sowing.

**Figure 2 viruses-14-01690-f002:**
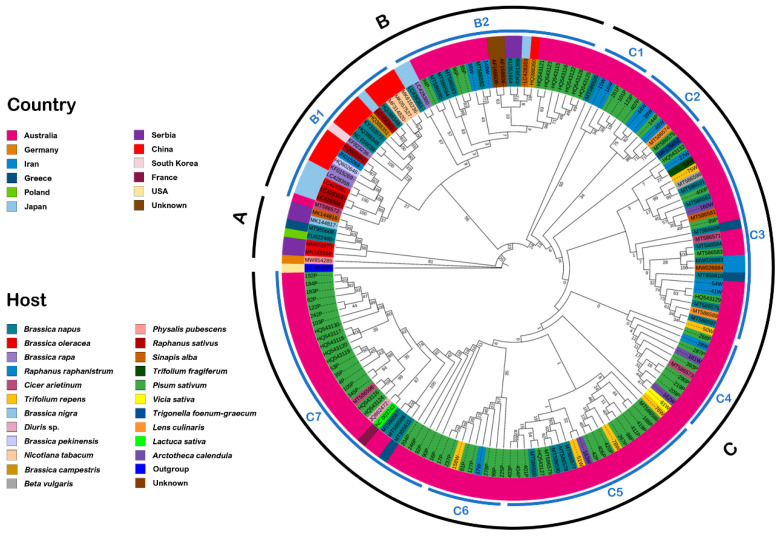
Phylogenetic analysis (maximum likelihood algorithm) based on nucleotide sequences of P0 of Tasmanian isolates. The phylogenetic trees show the evolutionary relationship of P0 of Tasmanian isolates with P0 of worldwide reported TuYV and BrYV isolates. The inner ring indicates the host, and the outer ring the country of origin.

**Figure 3 viruses-14-01690-f003:**
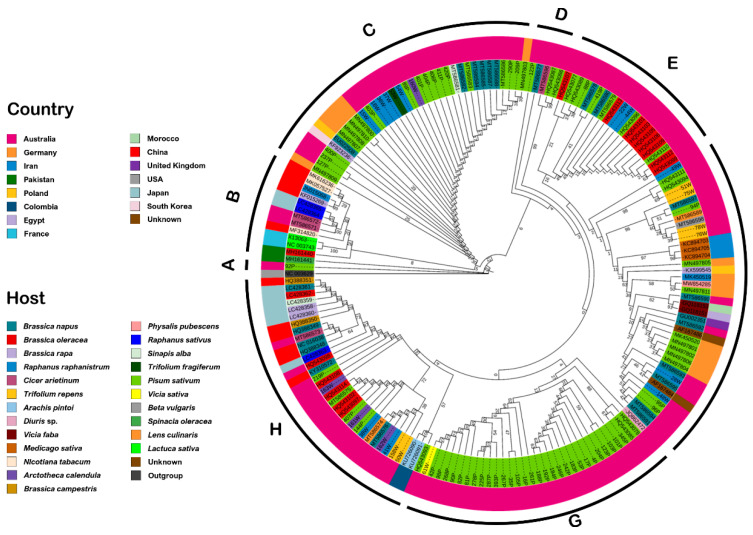
Phylogenetic analysis (maximum likelihood algorithm) based on nucleotide sequences of P3 of Tasmanian isolates. The phylogenetic trees show the evolutionary relationship of P3 of Tasmanian isolates with P3 of worldwide reported TuYV and BrYV isolates. The inner ring indicates the host, and the outer ring indicates the country of origin.

**Figure 4 viruses-14-01690-f004:**
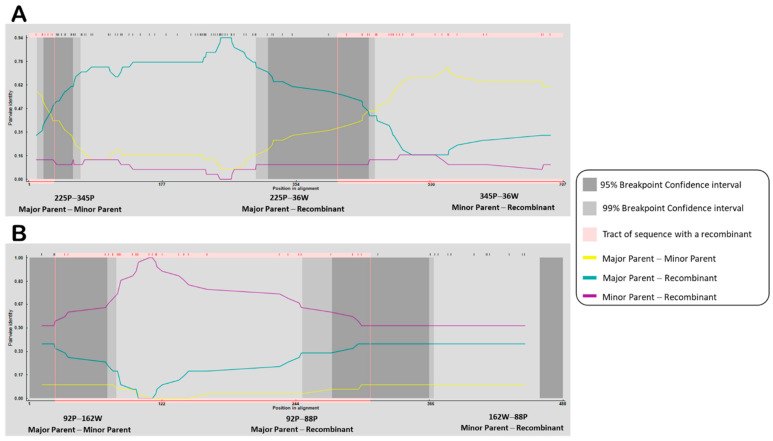
Results of the RDP recombination analysis for the (**A**) *P0* gene and (**B**) *P3* gene. Lines indicate the percentage of similarity per alignment.

**Figure 5 viruses-14-01690-f005:**
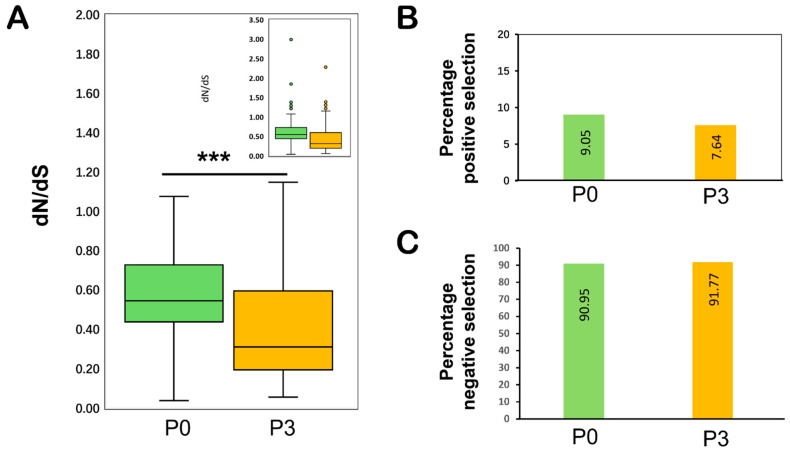
(**A**) Selection pressure was estimated by the calculation of non-synonymous to synonymous substitution ratios (dN/dS). Box plots correspond to dN/dS ratio for the TuYV-encoded *P0* gene and *P3* gene. The median values are represented by the horizontal lines inside the box, while inset shows the data with outliers indicated by small circles. Asterisks denote significance: *** *p* < 0.001. (**B**) Percentage of sites evolving under positive selection pressure in TuYV-encoded *P0* gene and *P3* gene. (**C**) Percentage of negatively selected sites among TuYV-encoded *P0* gene and *P3* gene.

**Table 1 viruses-14-01690-t001:** List of viruses recorded in Tasmania infecting green peas [4].

Species	Genus
*Alfalfa mosaic virus*	*Alfamovirus*
*Cucumber mosaic virus*	*Cucumovirus*
*Pea enation mosaic virus*	*Enamovirus*
*Phasey bean mild yellows virus*	*Polerovirus*
*Soybean dwarf virus*	*Luteovirus*
*Subterranean clover stunt virus*	*Nanovirus*
*Tomato spotted wilt virus*	*Orthotospovirus*
*Turnip yellows virus*	*Polerovirus*

**Table 2 viruses-14-01690-t002:** Primer sequences, genomic target, primer position, expected PCR product size and cycle conditions used in this study.

Primer Name	Sequence 5′ to 3′	Target	Position	Product Size (bp)	PCR Conditions	Reference
C2F1 + C2F2	TCACKTTCGGGCCGAGT	Luteoviridae (partial ORF3)		148	1 × 95 °C (15 min), 15 × [94 °C (30 s), 65 °C (30 s) reduce by 1 °C per cycle, 72 °C (30 s)], 25 × [94 °C (30 s), 50 °C (30 s), 72 °C (30 s)], 1 × 72 °C (7 min)	[40]
	TCACKTTCGGGCCGTCT					
C2R1 + 2	TCMAGYTCGTAAGCGATKG					
TuYVCP+	ATGAATACGGTCGTGGGTAGGAG	TuYV ORF3	3483	563	1 × 95 °C (15 min), 35 × [94 °C (30 s), 55 °C (30 s), 72 °C (1 min)], 1 × 72 °C (10 min)	[41]
TuYVCP−	CCAGCTATCGATGAAGAACCATTG		4045			
TuYVOrf0F	ACAAAAGAAACCAGGAGGGAATCCTTA	TuYV ORF0	1	780	As for TuYVCP+/TuYVCP− except 55 °C annealing	[29]
TuYVOrf0R	TCATACAAACATTTCGGTGTAGAC		781			
BWYVCPF	CAGTAGCCGGTATTTACTTAGTCTACC	BWYV ORF3	3472	648	As for TuYVCP+/TuYVCP− except 56 °C annealing	[3]
BWYVCPR	GGCACTTCATAGTGATTCTAAAAGAA		4119			
PhB7F	GATCCTTGTGCAAGTTTGTT	Partial 5′ UTR and 5′ end of ORF0	7	460	1 × 94 °C (1 min), 30 × [94 °C (30 s), 58 °C (60 s), 72 °C (3 min)], 1 × 72 °C (10 min)	[42,43]
PhB455R	GAATGAGACCTTTGTAAGTA		455			

**Table 3 viruses-14-01690-t003:** Prevalence of virus infections (*Luteoviridae* universal primers and species-specific primers) in weed species neighbouring pea crops.

Weed Species	Total Tested	Luteoviridae	SbDV	TuYV	BWYV	PBMYV	Undetermined	No Virus Detected
Unknown	7	0	0	0	0	0	0	7
*Arctotheca calendula*	4	4	0	4	0	0	0	0
*Medicago polymorpha*	6	0	0	0	0	0	0	6
*Rapistrum raphanistrum*	44	18	0	13	0	0	5	26
*Malva* sp.	6	1	0	0	0	0	1	5
*Trifolium pratense*	4	0	0	0	0	0	0	4
*Sonchus* sp.	13	2	0	1	0	0	1	11
*Trifolium fragiferum*	4	2	0	1	0	0	1	2
*Trifolium subterraneum*	13	1	0	0	0	0	1	12
*Vicia sativa*	9	5	3	1	0	0	1	4
*Trifolium repens*	54	33	21	9	0	0	3	21
**Total**	**164**	**66**	**24**	**29**	**0**	**0**	**13**	**98**

**Table 4 viruses-14-01690-t004:** Prevalence of virus infections (*Luteoviridae* universal primers and species-specific primers) in Tasmanian pea crops and TuYV incidence calculated using the Gibbs and Gower method (1960).

Site	U01	U02	U03	U04	U05	U07	U08	U10	U11	U12	U15	U16	U20	U22	U23	U26	U27	U32	U33	U34	U35	U36	Total
*Luteoviridae*	8	5	0	0	9	8	3	8	10	6	16	8	0	2	4	0	11	8	14	13	10	8	151
SbDV	0	0	0	0	0	1	0	0	0	0	0	1	0	0	0	0	0	1	2	0	0	0	5
BWYV	0	0	0	0	0	0	0	0	0	0	0	0	0	0	0	0	0	0	0	0	0	0	0
PBMYV	0	0	0	0	0	0	0	0	0	0	0	0	0	0	0	0	0	0	0	0	0	0	0
TuYV	8	5	0	0	9	7	3	8	10	6	16	6	0	2	4	0	11	7	6	11	10	8	137
TuYV incidence (%)	9.7	5.6	0.0	0.0	11.3	8.3	3.2	9.7	12.9	6.9	27.5	6.9	0.0	2.1	4.4	0.0	14.8	8.3	6.9	14.8	12.9	9.7	-
undetermined	0	0	0	0	0	0	0	0	0	0	0	1	0	0	0	0	0	0	6	2	0	0	9

Total number of samples tested is 440.

**Table 5 viruses-14-01690-t005:** Details of viruses detected through NGS.

	Viruses	Genus	Genome Length	Max Contigs Length (nt)	Type of Analysis	Samples ID
1	*Turnip yellows virus*	*Polerovirus*	5641 bp	1349, 5452, 457, 5588	Blastn	U1, U4, U5, U6
2	*Brassica yellows virus*	*Polerovirus*	5666 bp	3994	Blastn	U1,
3	*Turnip mosaic virus*	*Potyvirus*	9835 bp	9829	Blastn	U1,
4	*Raphanus sativus cryptic virus*	*unclassified Partitiviridae*	1866 bp	1700	Blastn	U1,
5	*Soybean dwarf virus*	*Luteovirus*	5853 bp	5842, 2919	Blastn	U2, U3
6	*White clover mosaic virus genome*	*Potexvirus*	5845 bp	1695, 352, 245	Blastn	U2, U3, U4
7	*Red clover vein mosaic virus*	*Carlavirus*	8604 bp	8631, 463	Blastn	U2, U3
8	*Clover yellow vein virus*	*Potyvirus*	9584 bp	9137	Blastn	U2,

**Table 6 viruses-14-01690-t006:** Recombination within and between *P0* and *P3* genes.

Analysed Region	Sequences Detected with Recomb. Event	Recombinant ^1^	Recombination Breakpoints “In Alignment (Without Gaps)”		Parental Sequences		Detection Methods ^2^	*p*-Value ^3^
			Begin	End	Major	Minor		
*P0*	5	36W	34 (32)	409 (390)	225P	345P	RMCS3	7.386 × 10^−3^
*P3*	34	88P	24 (23)	312 (310)	92P	162W	MS3	3.451 × 10^−7^

^1^ Numbering begins at the first nucleotide after the cleavage site at the replication origin and increases clockwise. ^2^ R, RDP; G, GeneConv; B, Bootscan; M, MaxChi; C, CHIMAERA; S, SisScan; 3, 3SEQ. ^3^ The described *p*-value corresponds to the program in bold, underlined type and is the lowest *p*-value calculated for the event in question.

**Table 7 viruses-14-01690-t007:** Molecular diversity within P0 and P3 sequences of TuYV isolates.

Dataset	Number of Sequences	Total Number of Sites	S	Eta	H	H_d_	π	*θ_w_*	Tajima’s D
P0	75	707	182	208	65	0.992	0.04829	0.05873	−0.61202
P3	75	488	97	110	42	0.948	0.04017	0.05046	−0.69341

S, number of polymorphic (segregating) sites; Eta, total number of mutations; H, number of haplotypes; H_d_, haplotype diversity; π, nucleotide diversity; *θ_w_*, Watterson’s theta.

**Table 8 viruses-14-01690-t008:** Estimation of average dN/dS ratios, positive and negative selection pressures within *P0* and *P3* genes of TuYV isolates.

ORF	Total Number of Codons	Avg. dN/dS Ratio	Positive Selection				Negative Selection			
			Total Sites	Avg.	Min.	Max.	Total Sites	Avg.	Min.	Max.
*P0*	222	0.6198	20	1.303	1.016	2.988	202	0.552	0.039	0.982
*P3*	158	0.4245	12	1.304	1.002	2.277	146	0.352	0.056	0.993

## Data Availability

The data is contained within the article or Appendix A.

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
