# Peer review of "Abundance of Poleroviruses within Tasmanian Pea Crops and Surrounding Weeds, and the Genetic Diversity of TuYV Isolates Found"

_viruses, 2022, doi:10.3390/v14081690_

Round 1
Reviewer 1 Report
In this article, the authors reported the prevalence and incidence of Polerovirus species (majority) and other viruses (minority) present in pea crops and neighbouring weeds that may act as virus reservoirs in Tasmania. They found that turnip yellows virus (TuYV) is the most prevalent polerovirus in all surveyed regions and analyzed the genetic diversity of the P0 and P3 genes of 75 Tasmanian TuYV isolates with worldwide reported TuYV isolates. The obtained results provide important information on epidemiological aspects and management of viral diseases of pea crops in Tasmania. Therefore, I recommend that it will be accept for publication in the Journal "Viruses"
As general comments, phylogenetic trees (Figures 2 and 3) of P0 and P3 are indistinct and unsuitable, and also difficult to understand for the readers. The authors mention that isolates used are divided into the several clades such as A, B, and C, but there is no indication at the figure, so please write clades at the figures (including supplemental figures). Also, bootstrap values should be indicated to be confident about the alignment being good. In others, description on data of NGS analysis is insufficient, so more detailed data would be shown in Table 5 (and Table S2).
My comments ands suggestions are given along the line numbers as follows.
Line 236: Table 3, it would be useful to write the number of "undetermined" for each species in additional column.
Line 239: Table 4, it would be useful to add "total number of positive samples". As well, the number of "undetermined" could be filled in.
Line 241: Please add " total numbers of samples tested are 440" in a footnote.
Line 243~: The authors give detailed numbers of reads for each sample in the text. The data such as no. of reads, no. of reads after trimming, no. of contigs, contig length etc. should be shown in Table S2. Also, sequence information (i.e., Accession number) of viruses detected from NGS analysis should be shown in either Table 5 or Table S2., because the authors give in detail viruses detected by NGS in Discussion (Line 484 to 522). Furthermore, it is interesting to know which samples (U1 to U6) contain any virus or which viruses are detected from any sample.
Line 296: Data on 22 distinct groups are not shown in Fig. 2. It should be indicated at the figure. More distinct tree should be devised.
Line 298~: The authors mention in detail host species and isolate numbers reported worldwide. To understand these relations, host species of each virus isolated from different countries should be added in virus isolate list of Table S3. References too, it would be useful.
Line 312~: "two major clades A and B, and sub-B clades" should be written in at the figure in Fig. 2.
Line 339: "three major clades A, B, and C" should be written in at the figure in Fig. S3.
Line 350: "clades A, B, and C" should be written in at the figure in Fig S4.
Line 366~: It would be useful to quote the respective references on different host plants for each isolate from different countries in Table S3.
Line 375: Ten major clades (A to J) should be written in at the figure in Fig. 3. More distinct tree should be devised.
Line 400: The authors regard as one event of recombination within P3 gene of 88P, but from data (MS3) of detection methods in Table 6 and figure of Fig. 4 A, it would be doubtful whether 88P is a recombinant.
Line 413: Fig. 4, please write isolate names with different colored lines in figure caption, because original drawing is unclear
Line 418~: Numerical data in parenthesis duplicate those of Table 7.
Line 425: Fig. 5 overlaps with data shown in Table 7, so Fig. 5 is not necessary.
Line 428: Table 7, "Total number of sites" should be added. It would be better for understanding it.
Line 446: Table 8, as with Table 7, "Total number of codons" is added. It would be helpful for understanding. P0 = 222; P3= 158.
Line 448: B and C of Fig. 6 are not necessary? Because the percentages of positive and negative selection can be easy to understand from Table 8.
Reviewer 2 Report
The manuscript could be improved by paying attention to the following issues:
It appears that the manuscript does not clearly show what the important information is. It might be due to too many information, or maybe because the data presentation is not very well organized. There are also many spots of concluding remarks without solid data presentation.
There are many abbreviations without appropriate explanation (PBMYV (line #141), BMYV, FBPolV, and BChV (lines #347-348)). The authors should describe what they stand for when they first appear. As for SbDV, it was explained when it first appeared in line #232. However, it was explained again what the SbDV was in line #252; ‘Soybean dwarf virus’ should be omitted in line #252.
Virus nomenclature in line #251-254 on page 8 is wrong and inconsistent.
There are many typographic errors (lines #5, #275, #293, #326, #394). Use of the word ‘whereas’ looks inappropriate in lines #409 and #532. Wrong use of comma (#286), and inappropriate omission of the comma (#258, #365, and #406) were also found.
Several grammatical errors were found in Discussion.
Round 2
Reviewer 1 Report
The MS has been improved, as suggested by the reviewers, but I have noticed the groupings of isolates shown in Figs. 2 and 3 and the terminology of TuYV and BrYV.
In this study, the P0 analysis shows that out of 75 Tasmanian TuYV isolates tested, 16 formed a clade with TuVY and 59 were closely related to BrYV, suggesting that most P0 sequences may be derived from BrYV more than from TuYV. It means that in Fig. 2, that is a main figure, Clade A groups clearly correspond to TuYV, whereas Clade B groups may correspond to BrYV rather than TuYV; nevertheless, the authors do not mention anything about this (also not shown in Fig. 2).
Likewise, the P3 analysis shows that all Tasmanian TuYV isolates were more closely related to TuYV, BrYV, BWYV, FBPolV, BMYV, and BChV (90.7−98.2% sequence similarities), suggesting that P3 sequences may originate from a complexed virus group that constitutes the at least six poleroviruses, probably due to recombination events of these viruses. Nevertheless, information of BrYV, BWYV, FBPolV, BMYV, and BChV are not included in groupings (A to J) in Fig. 3 that is a main figure.
More importantly, the fact that most P0 sequences (59/75) are more closely related with those of BrYV suggests that P1-P2 genes (core gens of poleroviruses) also are more closely related with those of BrYV. If this is correct, the term TuYV used in this study means TuYV/BrYV complex (indeed, BrYV can be thought to be a strain of TuYV). Such TuYV/BrYV relation can be confirmed from the NGS analysis, but is not be shown (in Table 5, BrYV is distinguished from TuYV). I wonder if the authors may have a misunderstanding (confusing?) on the TuYV and BrYV. In Discussion section, the authors do not discuss about TuYV and BrYV relations. Please consider these important, interesting points as well as the above-mentioned P3 complex, and so the aim of this study will become much more clear.
